# Improved Joint Health Following Oral Administration of Glycosaminoglycans with Native Type II Collagen in a Rabbit Model of Osteoarthritis

**DOI:** 10.3390/ani12111401

**Published:** 2022-05-30

**Authors:** Vicente Sifre, Carme Soler, Sergi Segarra, José Ignacio Redondo, Luis Doménech, Amadeo Ten-Esteve, Laura Vilalta, Luis Pardo-Marín, Claudio Iván Serra

**Affiliations:** 1Programa de Doctorado en Ciencias de la Vida y del Medio Natural, Escuela de Doctorado, Universidad Católica de Valencia San Vicente Mártir, 46002 Valencia, Spain; 2Hospital Veterinario UCV, Departamento de Medicina y Cirugía Animal, Facultad de Veterinaria y Ciencias Experimentales, Universidad Católica de Valencia San Vicente Mártir, 46002 Valencia, Spain; mdc.soler@ucv.es (C.S.); lauravilalta84@hotmail.com (L.V.); ci.serra@ucv.es (C.I.S.); 3Centro de Investigación Translacional San Alberto Magno, Universidad Católica de Valencia San Vicente Mártir, 46002 Valencia, Spain; 4R&D Bioiberica S.A.U., 08950 Esplugues de Llobregat, Spain; ssegarra@bioiberica.com; 5Departamento Medicina y Cirugía Animal, Universidad Cardenal Herrera-CEU, CEU Universities, 46115 Valencia, Spain; nacho@uchceu.es; 6Departamento de Matemáticas, Física y Ciencias Tecnológicas, Universidad Cardenal Herrera-CEU, CEU Universities, 46115 Valencia, Spain; luis.domenech@uchceu.es; 7Biomedical Imaging Research Group (GIBI230-PREBI), La Fe Health Research Institute, and Imaging La Fe node at Distributed Network for Biomedical Imaging (ReDIB) Unique Scientific and Technical Infrastructures (ICTS), 46026 Valencia, Spain; amadeogibi230@gmail.com; 8Interlab-UMU, Campus de Excelencia “Mare Nostrum”, University of Murcia, Campus Espinardo, 30071 Murcia, Spain; lpm1@um.es

**Keywords:** osteoarthritis, native type II collagen, glycosaminoglycans, DMOAD, SYSADOA, cartilage, subchondral bone, hyaluronic acid, MRI

## Abstract

**Simple Summary:**

Osteoarthritis is an incurable chronic disease. For this reason, new therapies are constantly emerging to improve clinical signs and the quality of life of our pets. Chondroitin sulfate, glucosamine and hyaluronic acid have been proven effective and are the most widely used in many formulations. In the present study, adding native type II collagen to the combination of chondroitin sulfate, glucosamine and hyaluronic acid showed improvements on osteoarthritis progression in an experimental model of osteoarthritis induced by transection of the cranial cruciate ligament of the knee in New Zealand white rabbits. Disease progression was monitored at different time points using magnetic resonance imaging biomarkers, measurement of hyaluronic acid in synovial fluid, and macroscopic and microscopic evaluations of cartilage, synovial membrane and subchondral bone. Overall, our results showed that adding native type II collagen to a combination of glycosaminoglycans allows a significantly slower osteoarthritis progression, compared to glycosaminoglycans alone.

**Abstract:**

A prospective, experimental, randomized, double blinded study was designed to evaluate the effects of glycosaminoglycans, with or without native type II collagen (NC), in an osteoarthritis model induced by cranial cruciate ligament transection. The following compounds were tested: chondroitin sulfate (CS), glucosamine hydrochloride (GlHCl), hyaluronic acid (HA) and NC. Fifty-four female 12-week-old New Zealand rabbits were classified into three groups: CTR (control–no treatment), CGH (CS + GlHCl + HA) and CGH-NC (CS + GlHCl + HA + NC). Each group was subdivided into three subgroups according to survival times of 24, 56 and 84 days. Over time, all rabbits developed degenerative changes associated with osteoarthritis. CGH-NC showed significantly improved values on macroscopic evaluation, compared to CTR and CGH. Microscopically, significantly better results were seen with CGH and CGH-NC, compared to CTR, and synovial membrane values were significantly better with CGH-NC compared to CGH. A significant improvement in magnetic resonance imaging biomarkers was also observed with CGH-NC in cartilage transversal relaxation time (T2) and subchondral bone D2D fractal dimension in the lateral condyle. In conclusion, our results show beneficial effects on joint health of CGH and CGH-NC and also supports that adding NC to CGH results in even greater efficacy.

## 1. Introduction

Osteoarthritis (OA) is one of the most common forms of arthritis. It is a chronic illness that correlates with pain and discomfort [1]. OA involves all tissues of the synovial joint including cartilage, subchondral bone, menisci and periarticular soft tissues, and its treatment has always been an orthopedic challenge due to the lack of an ideal treatment [2].

Non-steroidal anti-inflammatory drugs (NSAIDs) and other painkillers have traditionally been used in patients with OA, the treatment goal being to alleviate pain and minimize disability [3,4]. Glucosamine hydrochloride (GlHCl), chondroitin sulfate (CS) and hyaluronic acid (HA) are considered as symptomatic slow-acting drugs for OA (SYSADOA), and some of them have also shown a disease-modifying osteoarthritis drug (DMOAD) effect, as they show a delay on the progression of OA, reducing pain, stiffness, and joint swelling [5,6], as opposed to NSAIDs, which show less effects on OA progression [7].

CS is a sulfated GAG and one of the major components of the joint cartilage. Some effects of this GAG include acting as an anti-inflammatory, synthesis stimulation of proteoglycans and hyaluronic acid and inhibiting synthesis of proteolytic enzymes that cause cartilage matrix damage and death of chondrocytes [8,9]. Glucosamine is a monosaccharide found in the joint cartilage and is clinically effective in reducing pain and improving functionality and stiffness of the joint, delaying cartilage breakdown [10,11]. Combining CS with GS has been shown to provide similar effects on pain reduction, stiffness, joint swelling and effusion as some NSAIDs [5]. HA is also a GAG and a major component of the extracellular matrix. It has a high molecular weight and can be found in several animal fluids and tissues, such as synovial fluid, where it is produced by synoviocytes, fibroblasts and chondrocytes [12]. HA is commonly used for treatment of degenerative joint diseases, modulating arthritic pain and downregulating cytokines, free radicals and proteolytic enzymes in synovial fluid, preventing degradation of the articular cartilage [12,13]. Moreover, a recent publication reports improvements in OA biomarkers in dogs receiving oral HA after surgery for cranial cruciate ligament injury [14].

Over the past few years, administration of different types of collagens have been introduced, with these products showing a good effect compared to placebo groups [15,16]. In addition, a combination of GAGs with a low dose of native collagen type II (NC) can be effective as a pain reliever by mechanisms that involve protective effects on the cartilage as reported in a rat osteoarthritic model [17]. On top of this, NC simultaneously used with an NSAID, such as paracetamol, has revealed superior results than just using paracetamol on its own [18]. Nevertheless, other studies revealed that a combination of NC with GAGs leads to a lesser efficacy than GAGs alone [19].

Microscopic histological evaluation of OA has been performed over the past decades using different scales to evaluate cartilage and subchondral bone damage, such as those provided by the Osteoarthritis Research Society International (OARSI) [20,21,22]. Macroscopic evaluation has been also described by other authors, such as Tsurumoto et al. (2013) in evaluating osteophyte formation [23] or that described by Cook et al. (2010) evaluating the cartilage surface [24]. The rabbit cranial cruciate ligament transection (CCLT) OA model has shown good reproducibility and the ability to modify the gait [25] and achieve changes on the cartilage surface. In the rabbit model, these lesions are mainly identified on the lateral femoral condyle [26]. In addition to cartilage damage and osteophyte formation, the degree of synovitis and the synovial fluid concentration of HA are other parameters which can be evaluated and used in the synovial membrane inflammation score described by the OARSI [27] after quantifying the amount of hyaluronan in this fluid [13].

Among the less invasive methods available for evaluating OA, radiography has traditionally been one of the most commonly used techniques. Its main limitations are that only bone structures can be visualized, and that correlation with clinical signs is poor [28]. Conversely, magnetic resonance imaging (MRI) provides high resolution images of soft tissue joint structures, such as the articular cartilage. It also allows the use of imaging biomarkers, such as longitudinal relaxation time (T1) to quantify proteoglycan content [29] and transversal relaxation time (T2) to quantify water content and collagen structure [30], the latter being a risk factor for OA development [31]. In addition to articular cartilage, subchondral bone biomarkers can also be assessed by MRI as it means bone volume fraction (BV/TV), trabecular thickness (Tb.Th), trabecular spacing (Tb.Sp), trabecular number (TbN), quality of trabecular score (QTS) and D2D and D3D fractal dimension can be quantified. These parameters have been described using micro-computed tomography [32] and MRI scan [33,34].

Among the available animal models of OA, the induced model of anterior cruciate ligament section in rabbits has shown reproducibility, easy handling, and quick OA development [35], creating an unstable joint and leading to progression of the pathology [1]. Despite the anatomical and biomechanical differences between rabbits and humans [36], this model is reliable, creating changes in cartilage, subchondral bone and periarticular osteophytosis.

The objective of this study was to evaluate the effects on OA progression of a combination of CS, GlHCl and HA, with or without NC, for 84 days in a rabbit OA model. For this purpose, macroscopic and microscopic articular changes, synovial fluid HA concentrations and MRI cartilage and subchondral bone biomarkers were evaluated.

## 2. Materials and Methods

A prospective, experimental, randomized, and double-blinded study was designed. All procedures were performed according to the European legislation on protection of animals with the approval of the Local Animal Government Animal Protection Ethics Committee (RD53/2013).

### 2.1. Animal Model and Test Compounds

Fifty-four, twelve-week-old female New Zealand white rabbits were used in the study. The CCLT model was used to induce OA on their right stifle, while the left stifle was left as a non-diseased healthy joint.

The following compounds were used: CS (CS Bioactive^®^), GlHCl, HA (Mobilee^®^) and NC (Collavant n2^®^). All products and mixes were manufactured and provided by Bioiberica, S.A.U. (Esplugues de Llobregat, Spain). The animals were classified into three groups (*n* = 18) and received the following combinations of the above-mentioned products: CTR (control group–no treatment), CGH (60.38 mg/kg CS + 75.47 mg/kg GlHCl + 3.35 mg/kg HA) and CGH-NC (60.38 mg/kg CS + 75.47 mg/kg GlHCl + 3.35 mg/kg HA + 0.67 mg/kg NC). Each group was subdivided into three subgroups (*n* = 6) and, in each of these subgroups, rabbits were euthanized after 24, 56 or 84 days.

The day after CCLT surgery, oral administration of the assigned treatment was started. Compounds were given orally after diluting them in a 2-mL syringe with tap water. The treatment was administered by the same researcher, who was unaware of the composition of each of the different study treatments.

Once each subgroup reached their survival times, MRI was performed immediately after sacrifice. Following MRI, synovial fluid and synovial membrane samples were collected. Thereafter, stifles were photographed for macroscopic analysis, and both samples and stifles were labeled and preserved individually in a freezer at -80°C for further microscopic and biochemical analyses.

### 2.2. Histological Evaluation

Macroscopic evaluation

Following sacrifice, left and right stifles were dissected. By craniolateral approach to the skin, periarticular soft tissues dissection was performed. A 1.5-cm osteotomy proximal to the femoral trochlea and distal to the tibial plateau was done to retrieve the stifles. Once the stifle was isolated, periarticular soft tissues were meticulously dissected to retrieve the target biological samples. After removing all the stifle soft tissue, direct visualization of the joint was performed to score the samples following the macroscopic scale described by Laverty et al. (2010) [22] and the osteophyte stage described by Tsurumoto et al. (2013) [23].

b.Microscopic evaluation

Once anatomical samples were retrieved and after macroscopic evaluation had been performed, lateral femoral condyle and synovial membrane samples were extracted and fixed in formol. Following fixation with formaldehyde 4% and decalcification with EDTA (Osteodec^®^, LABOLAN, Navarra, Spain), these were included in 4-µm longitudinal section cuts of paraffin using a microtome. Femoral condyles were stained using hematoxylin and eosin and Masson’s Trichrome stain, and synovial membrane samples stained with hematoxylin and eosin. After staining, slides were digitalized for evaluation using a specific slide viewer software (CaseViewer 2.2^®^, 3DHISTECH Ltd., Budapest, Hungary).

Lastly, microscopic evaluation was done using the OARSI semi-quantitative scale described by Laverty et al. (2010) [22] to evaluate matrix stain, cartilage structure, chondrocyte density and cluster formation. The researchers also added an additional parameter, which was the outcome of adding the results of all measured variables as a total combined score. Structure of subchondral bone and disposition of cells in the synovial membrane were evaluated using semi-quantitative scales from OARSI described by Gerwin et al. (2010) [27].

### 2.3. Synovial Fluid Hyaluronic Acid Measurements

Synovial fluid HA extraction and analysis were performed following the methodology described by Liu et al. (2016) [37]. Lateral stifle arthrocentesis was performed to retrieve the synovial fluid by inoculating 1.5 mL of normal saline into the stifle joint. Approximately 10–15 movements of flexion and extension were performed to create a dilution of the synovial fluid to allow aspiration and preserved in Eppendorf tubes at −80 °C. Control samples of synovial fluid were taken from left stifles of a treatment group, prior to performing CCLT and administration of study treatments. HA measurement was performed using an ELISA test (TECO^®^ HA, TECOmedical, Headquarters, Sissach, Switzerland).

### 2.4. MRI Quantitative Biomarkers Analysis

MRI analysis was only performed in the CTR and CGH-NC group at survival time 84 days. Both stifles of each rabbit were analyzed considering the right stifles as osteoarthritis stifles (OA) and left as healthy stifles (Healthy).
Acquisition, preparation and processing of the imagesA 3 Tesla clinical scanner (Philips Achieva 3.0 TX, Madrid, Spain) with a 16-channel coil was used to perform the study. Cartilage imaging was performed with 3 different sequences on the sagittal plane for each individual stifle. A scan of the subchondral bone trabecula was performed with 3D high-resolution T1-weighted balanced Fast Field Echo (T1-FFE-3D) acquired on the transversal plane.Cartilage and subchondral bone images were transformed to NIfTI (Neuroimaging Informatics Technology Initiative) format to allow evaluation using the free distribution ITK-Snap software [38,39]. Femoral and tibial cartilage parcellation was done following the 6 segment scheme: medial anterior region (TM), lateral anterior region (TL), medial central region (CM), lateral central region (CL), medial posterior region (PM) and lateral posterior region (PL) [32,40] (Figure 1). Subchondral bone parcellation followed the 2-parcel scheme labeling both as medial and lateral [40].Prior to the cartilage image processing, open code Elastix toolbox [41]. was used for the spatial recording of the different eco times and flip angles into a common geometric space corresponding to a high-resolution Turbo Spin Echo T1 weighted sequence with fat suppression (T1 TSE SPIR). The imaging biomarkers were extracted using an ad-hoc program written in MATLAB (R2016b, Mathworks, Natick, MA, USA) for both cartilage and subchondral bone.Articular cartilage biomarkersCartilage longitudinal T1 relaxation time analysis was computed with the flip angles (2, 5, 10, 15, 25 and 45°) in a voxel-wise approach. The calculation for the longitudinal relaxation time was performed using the method described by Fram et al. (1987) [42,43].The cartilage transversal T2 relaxation time analysis used all the echo times (2.7, 4.1, 5.5, 6.9, 8.3, 9.7, 11.1, 12.5, 13.9, 15.3, 16.7, 18.1, 19.5, 20.9, 22.3, 23.7) and the method described by Li et al. (1996) [44].Cartilage volume and thickness analysis for each cartilage segments were obtained as described by Alberich-Bayarri et al. (2008) [33].Subchondral bone biomarkersTrabecular bone volume analysis used an algorithm based on local Laplacian to reduce heterogenicity and partial volume effect presents on the region of interest to obtain the bone volume fraction [45].Bone volume to total volume (BVTV) was calculated using the ratio between the number of voxels in the trabeculae and the total number of voxels of the volume of interest (VOI), Tb.Th and Tb.Sp were calculated based on the distance transformation of the skeleton on the contour as described by Alberich-Bayarri et al. (2008) [33]. TbN can be calculated as the ratio between BVTV and Tb.Th. The spatial distribution of the trabeculae was also evaluated by calculating the D2D and D3D Fractal Dimensions as described by Alberich-Bayarri et al. (2010) [34]. QTS was calculated, this biomarker provides a single score that reflects the quality of the bone trabecula (patent filing ID: 201931050)

### 2.5. Statistical Analysis

The statistical study was carried out using R statistical software version 4.1.1 (R Core Team, 2021). The effect size was calculated with pwr.t.test function from the pwr package (for groups with same sample sizes (*n* = 12). The alpha error was set at 0.05, the beta error was 0.8, and the alternative hypothesis was considered ”two-sided” [46]. An effect size of 1.2 was obtained. The normality of the studied variables was verified with a Shapiro-Wilk test. Homoscedasticity was checked using the Levene’s test. None of the studied variables met the normality and homoscedasticity criteria. For this reason, a robust approach was performed using the function t2way of the WRS2 package which computes a two-way ANOVA for trimmed means with interactions effects [47]. Statistical differences were considered when *p* < 0.05. The data are presented numerically as median, minimum to maximum, and they are graphically presented as median, inter-quartile range and minimum and maximum.

## 3. Results

### 3.1. Histological Study

#### 3.1.1. Macroscopic Evaluation

As expected when using this model, all rabbits developed degenerative changes associated with osteoarthritis after anterior cruciate ligament section. When treatment effect was evaluated together with survival time, CGH-NC was found to lead to significantly better macroscopic values, compared to CTR and also to CGH (*p* < 0.01), meaning that cartilage appearance in these rabbits was closer to that of a healthy one (Figure 2 and Figure 3).

#### 3.1.2. Microscopic Evaluation

Articular Cartilage

All microscopic cartilage variables except for “cluster formation” showed a significant worsening (*p* < 0.05) over time regardless of the treatment group. Only the “chondrocyte density” variable showed a statistically significant effect (*p* = 0.09) of CGH-NC over time, compared to the rest of the study groups (Figure 4 and Figure 6).

b.Subchondral Bone

A significant worsening (*p* < 0.05) with the survival time occurred regardless of the treatment group (Figure 5 and Figure 6).

c.Synovial membrane

Significantly better values (*p* < 0.01) were observed in the CGH-NC group, compared to CTR and CGH, regardless of the survival time, showing a less inflammatory stage. This variable was also influenced by the survival time regardless of the treatment group (*p* = 0.05) (Figure 7 and Figure 8).

### 3.2. Hyaluronic Acid

Synovial fluid analysis showed that CGH-NC administration was the only factor associated with changes in HA synovial fluid concentrations regardless of the survival time, leading to higher values (*p* = 0.07) (Figure 9) (Table 1).

### 3.3. MRI Imaging Biomarkers

#### 3.3.1. Articular Cartilage Biomarkers

MRI cartilage biomarkers revealed significant differences in nearly all variables between healthy and OA groups, regardless of the treatment. When treatments were compared, only significantly lower “T2 Femoral PL” values (*p* < 0.05) were observed with CGH-NC, compared to CTR. “T2 Tibial TM” compared to CTR group showed a *p* value of 0.07 (Figure 10).

#### 3.3.2. Subchondral Bone Biomarkers

Subchondral bone MRI biomarkers revealed significant differences in almost all studied variables between healthy and OA stifles regardless of the treatment group. Significantly better “D2D Femoral Lateral” values (*p* < 0.05) were achieved with CGH-NC, compared to CTR, while “D2D Tibial Lateral” values (*p* = 0.06) were also observed (Figure 11).

## 4. Discussion

The present study evaluates the potential effects of GAGs and NC on minimizing OA progression in a degenerative experimental CCLT-induced rabbit model. Results from this study reveal a beneficial effect of CGH, and that adding a low dose of NC to a combination of CS, GlHCl and HA leads to even greater efficacy. More specifically, CGH-NC showed significant improvements in macroscopic cartilage appearance, reduced synovial membrane inflammation and led to results closer to healthy joints on MRI biomarkers quantification.

Native type II collagen has shown efficacy in previous studies reducing OA pain in human medicine [15,48,49], as well as in veterinary medicine in species such as horses [19] and dogs [16,19,50,51]. In OA, native type II collagen acts through an oral tolerance mechanism of action, in a way that low doses achieve an immune-modulator effect that inhibits collagen type II destruction generated by T-cell response in the articular tissue [52]. This type of collagen needs to be differentiated from hydrolyzed collagen, which is also commonly used in the management of OA. Unlike native type II collagen, the effect of hydrolyzed collagen is based on oral administration at high doses aiming for absorption of peptide molecular sequences in the intestine, to then travel through the bloodstream and eventually reach the joint tissue [53,54].

Native type II collagen has been used in clinical and experimental studies as treatment for OA being referred as a symptomatic SYSADOA. Several publications support the benefits of NC Collavant n2^®^ in joint health [17,18,55], while other studies have been performed with other sources of NC [56,57]. The main symptomatic effect observed in these studies has been a decrease in pain, with an improvement on the joint range of motion, and a decrease on joint rigidity [50,58]. Likewise, studies using experimental models also have shown efficacy controlling OA pain [17]. However, Gupta et al. (2012) observed controversial results after combining NC with GlHCl and CS. They reported a lesser reduction of pain when combining these three compounds, compared to administering NC alone or to CS plus GlHCl [59]. It could be argued that these studies have some limitations, such as lack of homogeneity between groups initially regarding the degree of pain in the study by D´Altiglio et al. (2007) or unspecified the origin, molecular weight and dosage of the compounds used in Gupta et al. (2012) [50,59]. The disparity in the results from these different studies might also be explained by the NC being from a different source. Conversely, although the present study did not directly evaluate parameters related to articular pain, it did show a significant benefit of adding NC to CGH by decreasing the progression of joint degeneration. Simultaneously, an anti-inflammatory effect could be observed on the synovial membrane with the administration of GCH-NC.

The use of glycosaminoglycans for managing OA has been widely referenced. Although initially their use was controversial, new evidence has now proven a modulatory effect on joint degeneration, clinical pain improvement and a better function of OA joints [60]. The combination of CS plus GlHCl is the most commonly used and has shown a modulatory effect on OA progress as well as pain, function and joint rigidity at a clinical level [9,60]. The main effects seen with HA are joint lubrication [61], analgesic effect [62,63] and pro-inflammatory cytokine and metalloproteinase inhibition [64,65]. Traditionally, HA has been administered intraarticularly or parenterally [66]. However, lately, orally administered HA has been proven to lead to clinical [67,68] and OA synovial joint biomarkers [14] improvements.

There is a broad variety of in vivo OA models, which can be divided into two groups: spontaneous and induced. Spontaneous models are characterized by leading to a slower development of the disease. This is more representative of what occurs in OA progression, but it requires longer study periods. On the other hand, induced models, besides being able to develop of OA over a shorter period of time, they also allow to target a specific joint. In particular, CCLT is commonly used as induced OA model due to its reproducibility and similarities with OA processes in humans and dogs, creating the same changes and showing histological impact on the articular structures as early as after 4 weeks [35].

The present study further supports the validity of this model of CCLT-induced OA, as our data indicates that this method achieves an adequate induction of a degenerative OA process, as seen on most of the studied imaging and histological variables [1,2]. The key finding from the study, was the significant improvements obtained when adding NC to a combination of CS, GlHCl and HA. Such improvements could be seen at different levels; on the macroscopic evaluation, less erosive changes were seen on the cartilage surface; and on histology, with a reduced degree of synovitis, decreased synovial membrane hyperplasia and synoviocyte layers closer to a healthy-looking structure. These findings could be explained by the immunomodulatory effect of NC: modifying T-cell natural response in OA, destruction of collagen in the articular cartilage is minimized along with a controlled joint membrane inflammatory response [52].

Although, numerically, the present study showed an increased synovial fluid concentration of HA with CGH-NC, regardless of the survival time, no significant differences were found, unlike what occurred in prior studies [14]. Authors believe that the significance of the results seen in Serra et al. (2021) could be due to the resolution of the instability of the stifle, which did not occur in our study. The cranial cruciate ligament lesion was not repaired in our study, and this could lead to major or constant degenerative activity explaining why significant changes in HA synovial concentration were not seen.

MRI biomarkers quantification, in line with the outcomes from the macroscopic and microscopic evaluations, also showed a significant benefit after oral administration of CGH-NC. This combination achieved significant improvements in the articular cartilage transversal relaxation time (T2) and in the D2D fractal dimension subchondral bone biomarkers. T2 is sensitive to cartilage water content and anisotropic collagen type II organization. Variations in this biomarker are related with a preserved collagen structure which, in turn, could be explained by the NC mode of action [52,69]. D2D provides information on the fractal dimension in two dimensions, analyzing the organization of the subchondral bone trabecular structure. Changes observed in this biomarker with CGH-NC could be explained by a reduced mechanical stress in the subchondral bone trabeculae when weight is transferred through the joint preserving articular cartilage [33,34]. It is worthwhile mentioning that changes on imaging biomarkers (T2 and D2D) were seen on the lateral joint regions, specifically in the “T2 Femoral PL” and “D2D Femoral Lateral”. The authors believe that the following conclusions could be drawn: firstly, this is not a coincidence; it is known that the experimental model used in this study produces more mechanical stress in this compartment of the stifle [35]. In this study, an improvement in OA in these particular areas was seen macroscopically with CGH-NC and not with CGH or in the CTR. Secondly, imaging biomarkers evaluated with parcellation of the articular regions also confirmed that changes could be seen only in the lateral regions, being the remaining or the totality of the joint non-affected [32,40,70].

This study has some limitations which should be pointed out. First, although the addition of NC to CGH significantly improved a number of variables (macroscopic, synovial membrane, T2 femoral PL and D2D femoral lateral), some others were not affected. Authors suggest that this fact could be due to the short duration of the study itself, which might emulate an early OA stage with a medium-short evolution term (4, 8 and 12 weeks) rather than a longer one. Changes in this time period were perhaps insufficient to reveal significant degenerative changes in these variables conditioned by the treatment. This could be explained by 1) lack of sensitivity of some of the studied methods for early stages of OA, or 2) need for longer term evaluation in order to see an effect of these products on modification of disease progression [71]. Therefore, to obtain more conclusive results, longer term studies would be warranted. Second, no MRI biomarkers were quantified in the CGH group because CGH-NC and CTRL were prioritized, along with healthy stifles. Having this missing data would have helped in analyzing effects of the combined formula. Last, as the addition of NC to CGH is the key advantage seen in this study and also its main outcome, further studies should also evaluate the comparative effects of using NC Collavant n2^®^ alone, CGH and CGH-NC.

## 5. Conclusions

In conclusion, results obtained in this study show how the oral administration of CS with GlHCl and HA, with or without NC, is safe, and it provides significant improvements in OA progression using a rabbit CCLT-induced model. It also describes how adding NC to a combination of CS, GHCl and HA significantly increases its efficacy. More specifically, this addition leads to better outcomes seen on macroscopic and microscopic evaluation and MRI biomarkers. However, further prospective studies using MRI biomarkers are required to evaluate how different combinations of nutraceuticals modulate OA progression in longer survival times.

## Figures and Tables

**Figure 1 animals-12-01401-f001:**
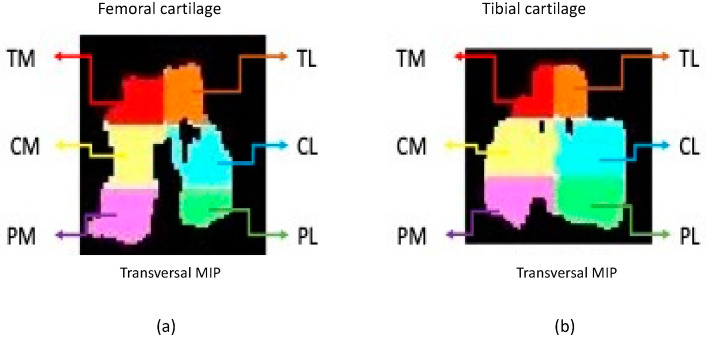
Transversal femoral (**a**) and tibial (**b**) cartilage parcellation over maximum intensity projection (MIP).

**Figure 2 animals-12-01401-f002:**
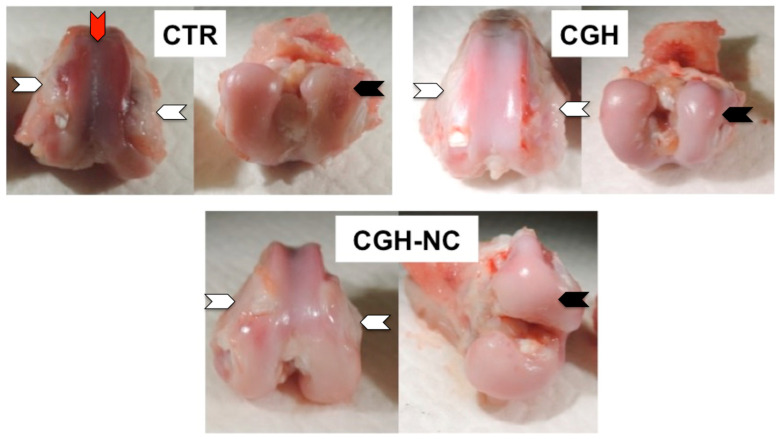
Macroscopic appearance of samples from the different study groups at 84 days. White arrows point to osteophytosis, black arrows point to ulcerations and the red arrow points to a cartilage eburnation.

**Figure 3 animals-12-01401-f003:**
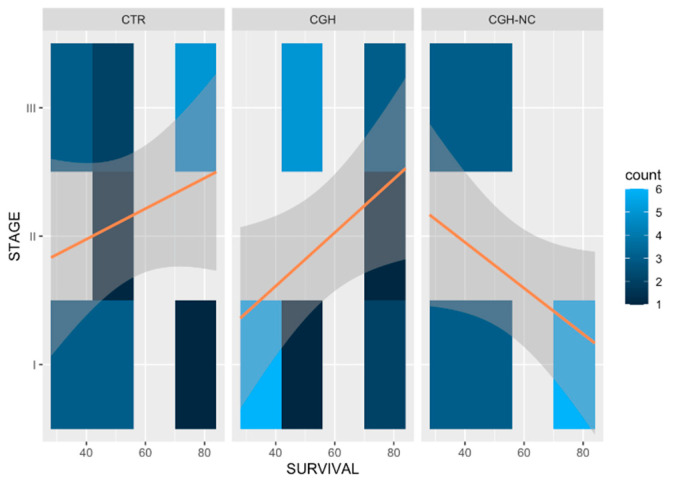
Heat maps representing changes in visual macroscopic stages for the different treatment groups and the frequency of occurrence at different survival times. Clearer colors result in an increase of the frequency of each stage of the disease. Orange lines are included on the maps to facilitate the understanding of the behavior of each treatment and the survival time. Significant differences (*p* < 0.05) can be appreciated in the CGH-NC group, compared to the rest of the treatment groups over time.

**Figure 4 animals-12-01401-f004:**
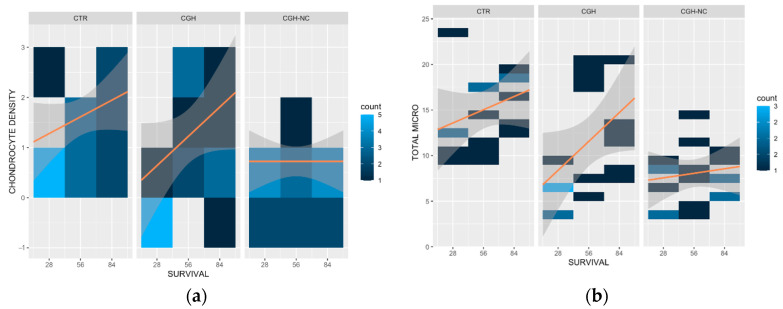
Heat maps representing “chondrocyte density” (**a**) and “total microscopic score” (**b**) for the different treatment groups and survival times and the frequency of occurrence at different survival times. Clearer colours result in an increase of the frequency of each stage of the disease. Orange lines are included on the maps to facilitate the understanding of the behavior of each treatment and the survival time. “Chondrocyte density” shows how CGH-NC lead to less degenerative changes when evaluated together with survival time (*p* < 0.1). “Total microscopic score” showed a statistically significant (*p* < 0.05) change with survival time regardless of the treatment.

**Figure 5 animals-12-01401-f005:**
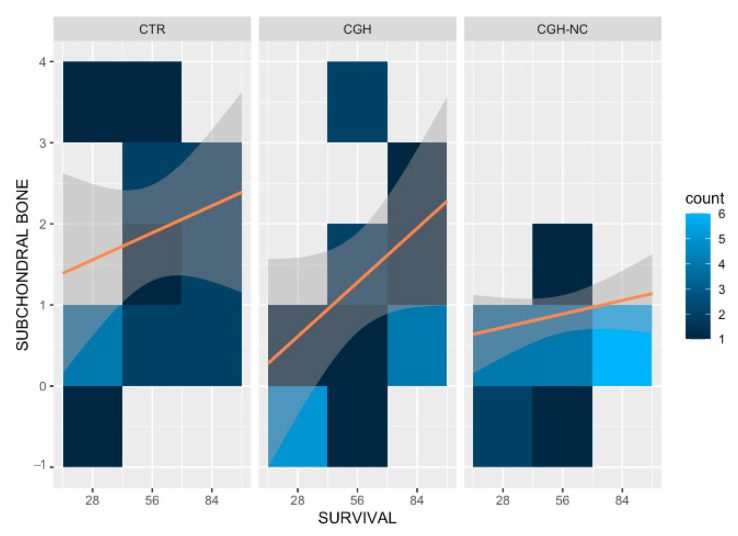
Heat map representing “subchondral bone” changes in the different treatment groups at different survival times and the frequency of occurrence at different survival times. Clearer colours result in an increase of the frequency of each stage of the disease. Orange lines are included on the maps to facilitate the understanding of the behavior of each treatment and the survival time. A statistically significant change occurred in all groups regardless of the treatment (*p* < 0.05).

**Figure 6 animals-12-01401-f006:**
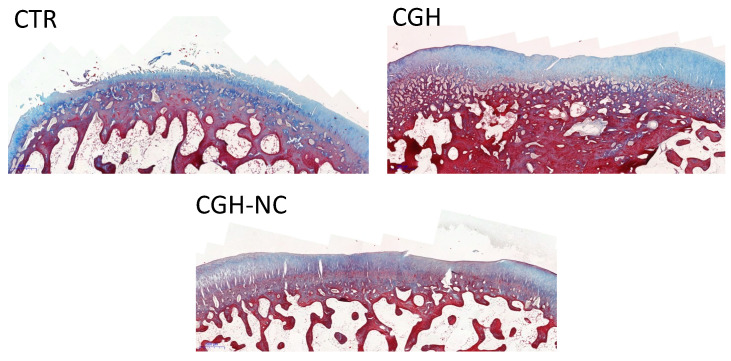
Histological sections of cartilage and subchondral bone (2×) using Masson’s trichrome stain of the different treatment groups. The different images show articular cartilage erosion, different extracellular matrix stains and subchondral bone structure changes.

**Figure 7 animals-12-01401-f007:**
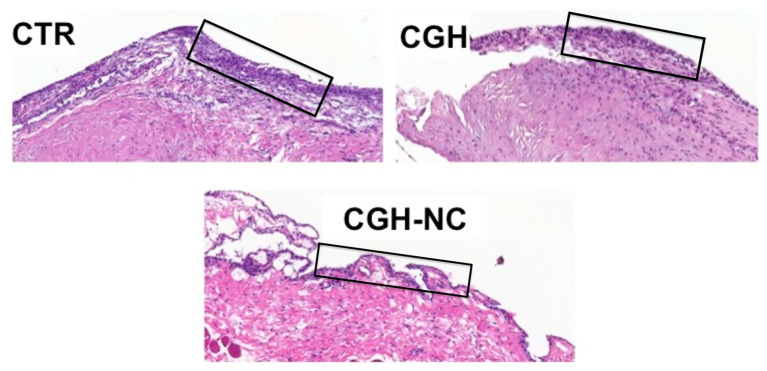
Synovial membrane histologic sections representative of each treatment group 84 days after surgery. Images from the CGH-NC group show a less hyperplastic appearance with only 2–3 synoviocyte layers, unlike the remaining groups. The black box shows the layers of sinoviocytes for each group.

**Figure 8 animals-12-01401-f008:**
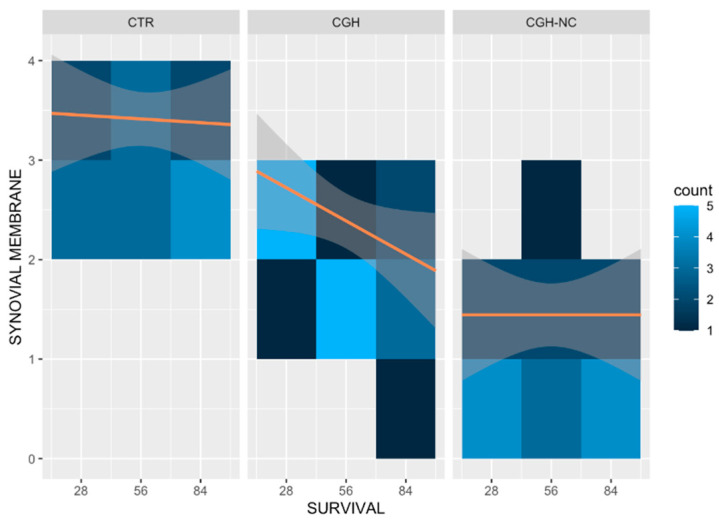
Heat map representing “synovial membrane” changes in the different treatment groups at different survival times and the frequency of occurrence at different survival times. Clearer colours result in an increase of the frequency of each stage of the disease. Orange lines are included on the maps to facilitate the understanding of the behavior of each treatment and the survival time. CGH-NC administration led to less inflammatory changes compared to the remaining treatment groups (*p* < 0.01).

**Figure 9 animals-12-01401-f009:**
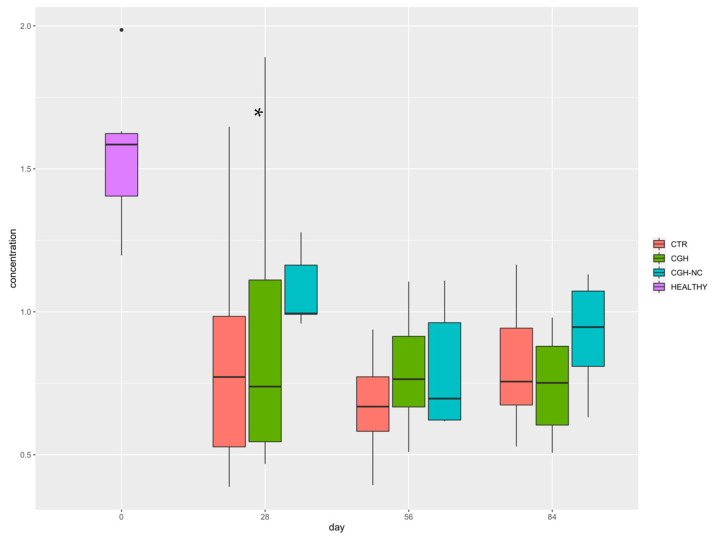
Hyaluronic acid concentration whiskers plot box for the three treatment groups at the three different survival times. * shows significant differences compared to the remaining groups (*p* < 0.05).

**Figure 10 animals-12-01401-f010:**
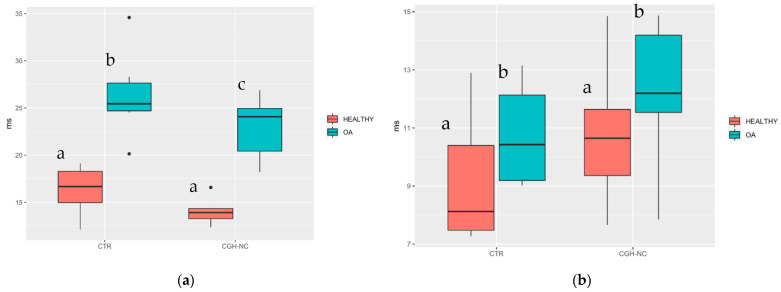
“T2 Femoral PL” (**a**) and “T2 Tibial TM” (**b**) whiskers plots boxes. Healthy stifles showed significant differences compared to OA stifles, regardless of the treatment group. Equally, in the “T2 Femoral PL” variable, differences can be seen in the OA stifles depending on the treatment administration. Different letters indicate statistically significant differences.

**Figure 11 animals-12-01401-f011:**
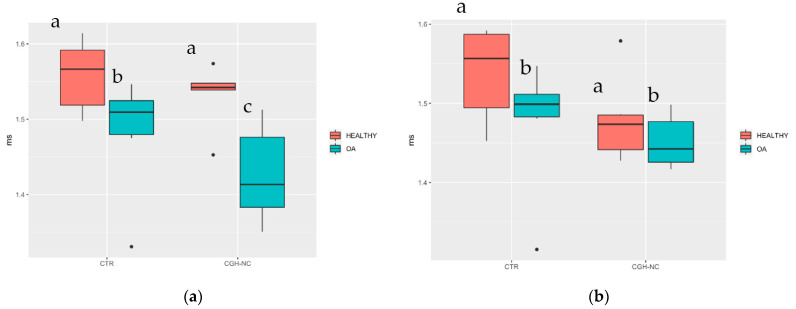
Whiskers plots box for “D2D Femoral Lateral” (**a**) and “D2D Tibial Lateral” (**b**) showing significant OA changes regardless of the treatment group. Equally, in the “D2D Femoral Lateral” variable, differences could be seen in the OA stifles between treatments. Groups with different letters refer to a statistical significance.

**Table 1 animals-12-01401-t001:** HA synovial fluid concentration ng/mL. (*p* < 0.05). Units are stated in ng/mL due to the HA concentration obtained following methodology used by Liu et al. (2016) [37].

	0 Days	28 Days	56 Days	84 Days
Healthy	243.30	-	-	-
CTR	-	53.21	48.46	71.54
CGH	-	61.35	67.85	59.52
CGH-NC	-	119.62	68.57	90.66

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
