# Peer review of "Improved Joint Health Following Oral Administration of Glycosaminoglycans with Native Type II Collagen in a Rabbit Model of Osteoarthritis"

_animals, 2022, doi:10.3390/ani12111401_

Round 1
Reviewer 1 Report
This is an interesting study about improved joint health following oral administration of glycosaminoglycans with native type II collagen in a rabbit model of osteoarthritis. However, the following points should be addressed before it could be considered for the publication in Animals.
- Line 27, glucosamine is an amino sugar (monosaccharide), but not glycosaminoglycan, which is long linear polysaccharide consisting of repeating disaccharide units. Line 62-63, glucosamine hydrochloride (GlHCl) is not glycosaminoglycan for the same reason.
- There are different methods to induce OA in the animal model. The authors should add the discussion about why CCLT model was used to induce OA in this study.
- Histologic sections of the cartilages with different treatments should be added.
- Line 404, 'The main effects seen with HA are joint lubrication...' One recent study related to the role of HA in joint lubrication (Cells 9 (7), 1606) should be included.
- Table 1, the HA concentration in synovial fluids is in the order of mg/mL, not ng/mL level.
- Collagen 2 is a protein, which accounts 85–90% of collagen of articular cartilage. However, oral administration was given for the OA animals, which would be digested on the way to joints. The authors should explain why oral administration of collagen 2 would benefit the recover of OA.
Author Response
In reply to your comments:
- Glycosaminoglycans has been removed. Line 26 and line 64
-
Paragraph added on line 465-472
- Histological sections for the three treatments are added as figure 6
-
Reference added to manuscript and references, line 460 and lines 725-726
-
Hyaluronic acid concentration changed to mg/ml
-
The second paragraph of the discussion mentions that the objective of native type II collagen, when administered orally at low doses, targets an immune-modulator effect on the T-cell response inhibiting collagen type II destruction. This effect is the main difference compared to hydrolyzed collagen which requires high oral doses to promote intestinal absorption and further distribution to the articular tissue.
Modification of “NC” to native type II collagen has been done for better understanding on lines 429 and 433.
Reviewer 2 Report
Dear Sirs,
this is an interesting, well-written paper that adds useful data to the literature.
Apart from one sentence that needs to be rephrased, I have no other comments.
Lines 157-158: “Using a craniolateral approach of the skin and dissection of the soft tissues around the femur and tibia were performed”. Unclear. Please rephrase
Author Response
Sentence changed for better understanding, line 160-161.
Reviewer 3 Report
Authors presented a very interesting in vivo study to analyse the improvement in osteoarthritis progression adding different substances (native type II collagen, glucosamine and hyaluronic acid). They have demonstrated that adding native type II collagen combined with glycosaminoglycans slows the progression of osteoarthritis.
I have few comments:
- Authors indicated that the study is double-blinded. I do not see how this was implemented. It is said that the researcher administrating the treatment was unaware of the composition.
- Authors should explain the heat maps (Figures 3, 4, 5 and 7). It is difficult to understand the results.
- Figure 8-10 should include if the results are statistically different (using * for example), and indicate the meaning of the dots.
- Authors should add future improvements or future research lines to continue this study.
Author Response
- Authors indicated that the study is double-blinded. I do not see how this was implemented. It is said that the researcher administrating the treatment was unaware of the composition.
The researcher administering the treatments did not know the formula and concentrations of them. There were a formula A, B and C. Until the results were not obtained the researchers did not know the composition for each of the groups.
- Authors should explain the heat maps (Figures 3, 4, 5 and 7). It is difficult to understand the results.
Heat maps have been updated for a better understanding
- Figure 8-10 should include if the results are statistically different (using * for example), and indicate the meaning of the dots.
* have been used to indicate statistical significance. Figures 9-11
- Authors should add future improvements or future research lines to continue this study.
we have added the following sentence in line 539-544.
“However, further prospective studies using MRI biomarkers are required to evaluate how different combinations of nutraceuticals modulate OA progression in longer survival times”.
Round 2
Reviewer 1 Report
While the authors sovled my most of the comments in the revision, the unit of HA concentration in the synovial fluid is still problematic.
In table 1, the unit of HA is changed to mg/mL based on my requirement. However, the number is in the order of 10-5, which means it is in the ng/mL range. It doesn't make sense at all since the HA concentration in human SF ranges from 1 to 4 mg/mL (the one in rabbit SF would be in the range of ng/mL or tens ng/mL).
Author Response
To obtain synovial liquid and calculate HA concentration, methodology described by Lui et al. (2016) has been followed. This methodology performs an articular lavage of 1.5ml of sterile saline inside the stifle joint to be able to retrieve the sample in species of this size like rabbits. The reason of expressing the results in this units is because we can´t extrapolate the results to mg/ml because the dilution occurs inside the stifle and each animal will have different concentrations of synovial fluid.
Table units have been changed to inform about methodology used.